# Cubic AgBiS_2_ Powder Prepared Using a Facile Reflux Method for Photocatalytic Degradation of Dyes

**DOI:** 10.3390/mi14122211

**Published:** 2023-12-07

**Authors:** Wenzhen Wang, Chengxiong Gao, Yuxing Chen, Tao Shen, Mingrong Dong, Bo Yao, Yan Zhu

**Affiliations:** 1Shanghai Technical Institute of Electronics & Information, Shanghai 201411, China; wzwang@t.shu.edu.cn; 2School of Materials Science and Engineering, Shanghai University, Shanghai 200444, China; 3Faculty of Materials Science and Engineering, Kunming University of Science and Technology, Kunming 650093, China; gaocx2017@163.com (C.G.); shentao@kust.edu.cn (T.S.);; 4Zhejiang Engineering Research Center of MEMS, Shaoxing University, Shaoxing 312000, China; sc01240218@outlook.com (Y.C.); yaob_usx@163.com (B.Y.)

**Keywords:** AgBiS_2_, reflux method, photocatalytic degradation

## Abstract

The ternary chalcogenide AgBiS_2_ has attracted widespread attention in the field of photovoltaic and photoelectric devices due to its excellent properties. In this study, AgBiS_2_ powders with an average diameter of 200 nm were prepared via a simple and convenient reflux method from silver acetate, bismuth nitrate pentahydrate, and n-dodecyl mercaptan. The adjustment of the ratios of Ag:Bi:S raw materials and of the reaction temperatures were carried out to investigate the significance of the synthesis conditions toward the composition of the as-synthesized AgBiS_2_. The results of XRD indicated that the powders synthesized at a ratio of 1.05:1:2.1 and a synthesis temperature of 225 °C have the lowest bismuth content and the highest purity. The synthesized AgBiS_2_ crystallizes in a rock salt type structure with the cubic $Fm\bar{3}m$ space group. Scanning and transmission electron microscopy, thermogravimetric analysis, ultraviolet–visible–near-infrared spectra, and photocatalytic degradation performance were employed to characterize the as-synthesized samples. The results demonstrated that AgBiS_2_ powders display thermal stability; strong absorption in the ultraviolet, visible, and partial infrared regions; and an optical bandgap of 0.98 eV. The obtained AgBiS_2_ powders also have a good degradation effect on the methylene blue solution with a degradation efficiency of 58.61% and a rate constant of 0.0034 min^−1^, indicating that it is an efficient strategy for sewage degradation to reduce water pollution.

## 1. Introduction

With the rapid development of the industry, chemical plants continue to discharge organic wastewater, leading to increasingly severe water pollution. This pollution has a serious impact on the water used in human daily life and the environment and poses a significant threat to human health. Therefore, it is crucial to find a simple, economical, and efficient method for sewage degradation to reduce pollution and achieve sustainable development. The traditional sewage treatment methods include precipitation, ion exchange, membrane separation, and adsorption, etc., but these methods have the disadvantages of high cost, low selectivity, and serious secondary pollution, which limits their practical application. In recent years, the photocatalytic degradation of organic matter by semiconductor materials has attracted much attention due to its simplicity, high efficiency, low energy consumption, and environmental friendliness.

The ternary chalcogenide semiconductor material AgBiS_2_ exhibits unique properties, such as a high absorption coefficients and environmental stability, making it an attractive material for various applications [1]. It belongs to the I–V–VI_2_ family of compounds (where I = Cu/Ag/Au; V = As/Sb/Bi; and VI = S/Se/Te) [2], which are composed of earth-abundant, non-toxic elements. It has been proven that AgBiS_2_ has potential in fields like the photovoltaic, photoelectric, thermoelectric, and so on, but there are few reports focusing on its ability for photocatalytic degradation. Ganguly et al. [3] synthesized the AgBiS_2_-TiO_2_ composite to photodegrade doxycycline, and the degradation effect exceeded 95% within 180 min, which demonstrated the photocatalytic degradation ability of AgBiS_2_.

The history of the synthesis and the crystal growth of the ternary chalcogenide semiconductor material AgBiS_2_ can be traced to the first report by Wernick [4,5] in 1959. He reported a complete series of AgBiSe_2_/AgBiS_2_ of β (low temperature, hexagonal) and α (high temperature, cubic) solid solutions with transformation temperatures ranging from 200 to 300 °C [4]. Since then, various methods have been reported for preparing AgBiS_2_, such as solvothermal [6,7,8], hot-injection solution [9,10,11], hydrothermal synthesis [12], sonochemical [13], microwave [14], colloidal [15], solution-phase ligand-exchange [16], and a solid phase sintering method [17], among which the solvothermal and hot-injection solution methods are the most commonly used. Manimozhi et al. [6] synthesized AgBiS_2_ nano-/microspheres through the solvothermal method using ethylene glycol as the solvent and polyethylene glycol as a soft template at 180 °C for 20 h. Thongtem et al. [7] produced and purified AgBiS_2_ nanostructured flowers from CH_3_COOAg, Bi(NO_3_)_3_·5H_2_O, and thiosemicarbazide using different solvents in Teflon-lined stainless steel autoclaves via the solvothermal method at the reaction temperature of 200 °C for 24 h. Bernechea et al. [9] developed a hot-injection synthetic route for the synthesis of colloidal AgBiS_2_ nanocrystals in an Ar atmosphere. The significance of the composition and sensitivity toward synthesis conditions of the hot-injection synthesis method was investigated by adjusting the concentrations and ratios of Ag and Bi precursors, as reported by Öberg et al. [10]. Kaowphong [12] successfully prepared AgBiS_2_ nanostructures via biomolecule-assisted hydrothermal synthesis at 200 °C for 12–72 h. Pejova et al. [13] synthesized 3D arrays of close-packed AgBiS_2_ quantum dots (QDs) in thin film form using the sonochemical approach with an annealing treatment in an air atmosphere at 250 °C. Cubic AgBiS_2_ nanoparticles and flower-like clusters were successfully synthesized via 100 W and 800 W microwave refluxing in ethylene glycol for 60 min [14]. Bellal et al. [17] investigated the physical and photoelectrochemical properties of α-AgBiS_2_, synthesized from Bi_2_S_3_ and Ag_2_S in an evacuated Pyrex ampoule at 550 °C.

Although all of the aforementioned methods can synthesize high-quality AgBiS_2_, they also require complex equipment to achieve high temperatures or high pressures, and the reaction times are usually over 12 h. In this study, high-quality AgBiS_2_ powders were synthesized using a simple and convenient reflux method, and their physical performance and the photocatalytic degradation of the methylene blue solution under the simulated sunlight were studied.

## 2. Materials and Methods

### 2.1. Materials

Silver acetate (CH_3_COOAg; Shandong West Asia Chemical Industry Co., Ltd., Linyi, China), bismuth nitrate pentahydrate (Bi(NO_3_)_3_·5H_2_O; Tianjin Kemiou Chemical Reagent Co., Ltd., Tianjin, China), thiourea (SC(NH_2_)_2_; Tianjin Fengchuan Chemical Reagent Technology Co., Ltd., Tianjin, China), n-dodecyl mercaptan (C_12_H_25_SH; Aladdin Reagent Co., Ltd., Shanghai, China), methanol (CH_3_OH; Tianjin Fengchuan Chemical Reagent Technology Co., Ltd., Tianjin, China), n-hexane (C_6_H_14_; Tianjin Fengchuan Chemical Reagent Technology Co., Ltd., Tianjin, China), Ethylene glycol ((CH_2_OH)_2_; Tianjin Fengchuan Chemical Reagent Technology Co., Ltd., Tianjin, China), methylene blue (C_16_H_18_ClN_3_S; Tianjin Kemie Ou Chemical Reagent Co., Ltd., Tianjin, China) were used.

### 2.2. Preparation of AgBiS_2_ by the Reflux Method

Silver acetate and bismuth nitrate pentahydrate were dissolved in 30 mL of n-dodecyl mercaptan solution in a three-port flask. The mixture was stirred with a magnetic stirrer and aided by ultrasonic shaking to promote dissolution. The molar ratios of the silver source, bismuth source, and sulfur source were set at 1:1:2, 1.05:1:2.1, 1.1:1:2.2, and 1.15:1:2.3, respectively. Thiourea was then added to this orange solution. Subsequently, a condenser tube and a vacuum pump were connected to the completely dissolved solution, and the solution was vacuumed at 130 °C for 5 min. Vacuuming was then stopped, and the temperature was raised to 175 °C and held for 30 min, followed by heating to the desired reaction temperature and maintaining it for an hour. The reaction temperatures were set at 175, 200, 225, 250, and 275 °C, respectively. After the completion of the reaction, the solution was naturally cooled to about 60 °C, and the solution was poured into the mixed solution consisting of methanol and anhydrous ethanol with a volume ratio of 1:1 in a beaker. The solids on the wall of the three-port flask were cleaned with n-hexane and then poured into the beaker. A significant number of black solids were obtained after centrifuging this mixed solution. Finally, the black solids were washed several times with anhydrous ethanol and ultrapure water. After cleaning, the AgBiS_2_ black solids were then dried in a blast drying oven at 60 °C for 2 h to obtain AgBiS_2_ powders.

### 2.3. Photocatalytic Degradation Experiment

The 40 mg catalyst was added to the prepared methylene blue solution at a concentration of 5 mg/L. The mixture was stirred in a dark environment for one hour to ensure the adsorption–analytical equilibrium between the catalyst and the methylene blue molecules. The solution was then divided into two parts, one of which was filtered to remove the catalyst, leaving only the methylene blue solution to be tested, named Solution A, and the other solution containing the catalyst, named Solution B. After that, Solution A and B were placed under a 400 W high-pressure sodium lamp to simulate sunlight for the photocatalytic degradation experiment to ensure that the distance between each sample and the light source was the same and to eliminate the influence of the light source intensity on the experiment. After 1, 2, 3, and 4 h of simulated light irradiation, a 4 mL solution was extracted from Solution B, the methylene blue solution was left after removing the catalyst via filtration, and the 4 mL solution was also extracted from Solution A as the blank control sample, respectively. The light absorption intensity of the methylene blue solution was determined using a UV–visible near-infrared spectrophotometer (U-4100).

### 2.4. Characterization

The structure and phase purity of the synthesized powders were characterized using X-ray diffraction (XRD) performed on a DX-2700BH instrument (Haoyuan Instrument, Dandong, China). The XRD analysis utilized Cu Kα radiation (λ = 0.154 nm) in the 2θ range of 5–90° and an increment of 0.026° and operated at 40 kV and 30 mA. The morphology of the samples was recorded via ultra-high resolution field emission scanning electron microscopy (FE-SEM) (SU8020, Hitachi SU8000 series, Hitachi City, Japan) operated at 3 kV. The light absorption intensity of samples were characterized using the ultraviolet–visible–near-infrared spectrophotometer (U-4100, Hitachi, Hitachi City, Japan) with a scan speed of 100 nm min^−1^ and a slit width of 2 nm within the wavelength range of 200–1400 nm at room temperature. A highly refined barium sulfate (BaSO_4_) plate was used as the reference. The chemical elements in the samples were detected using a 300 kV field emission transmission electron microscope (FE-TEM) (Tecnai G2 F30 S-TWIN, FEI, Hillsboro, OR, USA) in a multi-functional, multi-user environment. The thermal stability and composition of the AgBiS_2_ powders were measured via thermogravimetric (TG) analysis (TG-DTA8122, Rigaku, Akishima, Japan).

## 3. Results and Discussion

### 3.1. Structural Characteristics

As shown in Figure 1, the XRD patterns were employed to investigate the influence of different reaction temperatures on AgBiS_2_ powders prepared using the reflux method with a raw material ratio of Ag:Bi:S of 1:1:2. The cubic AgBiS_2_ (PDF# 89-2045) was synthesized at different reaction temperatures, which resulted a face-centered statistically NaCl-type [5] structure with the cubic $Fm\bar{3}m$ space group and a unit-cell parameter of 5.648 Å (Figure 2a). In the cubic AgBiS_2_, Ag and Bi atoms are distributed indistinguishably and on the average in one set of positions (0, 0, 0; 0, ½, ½; ↻), and the S atoms are distributed on the average in the other set of positions (½, ½, ½; 0, 0, ½; ↻) [5]. The diffraction angles of the XRD patterns 2θ = 27.327°, 31.657°, 45.380°, 53.786°, 56.385°, 66.121°, 72.950°, 75.167°, and 83.846° correspond to the AgBiS_2_ diffraction crystal planes (111), (200), (220), (311), (222), (400), (331), (420), and (422), respectively. The remainder of the diffraction peaks are consistent with the characteristic peaks of bismuth, indicating the presence of Bi as the only impurity in the product. The XRD pattern at 175 °C shows the characteristic peak of AgBiS_2_, indicating that the synthesis temperature of AgBiS_2_ is below this temperature, which is consistent with previous reports. Using silver sulfide and bismuth sulfide as raw materials, Bellal et al. [17] concluded that AgBiS_2_ was endothermic when formed at 173 °C. As shown in Figure 2b, the binary phase diagram of Ag_2_S-Bi_2_S_3_ also supports the formation of AgBiS_2_ after exceeding 175 °C [18,19,20]. It is worth noting that, except for the XRD pattern at the reaction temperature of 225 °C, the peak intensities at 2θ = 27.327° are higher than that at 2θ = 31.657°, which is due to the mechanical superposition of the characteristic peaks of AgBiS_2_ and Bi. This result indicates that the reaction yield is the highest and the proportion of raw material wastage is the smallest at 225 °C. Therefore, based on these results, a synthesis temperature of 225 °C was used for further experiments.

To purify the AgBiS_2_ product and minimize the formation of the secondary phase, subsequent experiments were performed by increasing the molar amounts of silver and sulfur sources. The initial molar ratios of raw materials of Ag:Bi:S were set at 1:1:2, 1.05:1:2.1, 1.1:1:2.2, and 1.15:1:2.3, respectively. Figure 3 shows the XRD patterns of the synthesized products with different molar ratios at 225 °C. With the increase in silver and sulfur content, the characteristic peaks of bismuth are weakened, but they do not disappear when the contents of silver and sulfur continue to increase. This is because the generation temperature of bismuth is lower than that of AgBiS_2_, resulting in bismuth nucleation occurring prior to AgBiS_2_ nucleation. Therefore, the weak characteristic peak of bismuth still exists in the spectrum. It is worth noting that the ratio of the peak intensity of the characteristic peak of Bi at 2θ = 37.954° to that of AgBiS_2_ at 2θ = 31.657° is the smallest in the XRD spectrum of AgBiS_2_ synthesized with a ratio of raw materials of 1.05:1:2.1. This proves that the powder synthesized with this ratio has the lowest bismuth content and the highest purity. Therefore, the raw material ratio of 1.05:1:2.1 was selected for the preparation of AgBiS_2_ in the subsequent experiments.

In the process of powder preparation via the reflux method, a yellow emulsion was formed when the silver source and bismuth source were added to the solvent. This occurrence arises from the ultrasonic dissolution of silver acetate and bismuth nitrate pentahydrate in n-dodecyl mercaptan (NDM), resulting in the formation of light yellow [Ag(C_12_H_25_SH)_m_]^+^ (Ag-NDM) turbid liquid and yellow [Bi(C_12_H_25_SH)_n_]^3+^ (Bi-NDM). Subsequently, upon the addition of thiourea (TU), an orange substance formed, indicating a reaction of the solution system within thiourea. The color of the solution system turns black after gradually heating up, and finally, black AgBiS_2_ particles are generated. It is worth noting that, thiourea in the reaction mixture not only plays a role in forming a complex but also as the source of sulfur. All the reactions are expressed via the following equations:(1)$${Ag}^{+}+m\;C_{12}H_{25}SH\to {\left[Ag{\left(C_{12}H_{25}SH\right)}_{m}\right]}^{+}$$
(2)$${Bi}^{3+}+{n\;C}_{12}H_{25}SH\to {\left[Bi{\left(C_{12}H_{25}SH\right)}_{n}\right]}^{3+}$$
(3)$${\left[Ag{\left(C_{12}H_{25}SH\right)}_{m}\right]}^{+}+m\;SC{\left(NH_{2}\right)}_{2}\to {\left\{Ag{\left[SC{\left(NH_{2}\right)}_{2}\right]}_{m}\right\}}^{+}+m\;C_{12}H_{25}SH$$
(4)$${\left[Bi{\left(C_{12}H_{25}SH\right)}_{n}\right]}^{3+}+n\;SC{\left(NH_{2}\right)}_{2}\to {\left\{Bi{\left[SC{\left(NH_{2}\right)}_{2}\right]}_{n}\right\}}^{3+}+n\;C_{12}H_{25}SH$$
(5)$${\left\{Ag{\left[SC{\left(NH_{2}\right)}_{2}\right]}_{m}\right\}}^{+}+{\left\{Bi{\left[SC{\left(NH_{2}\right)}_{2}\right]}_{n}\right\}}^{3+}\to AgBiS_{2}+\left(m+n-2\right)\;SC{\left(NH_{2}\right)}_{2}+2CNNH_{2}+4H^{+}$$

At the beginning of the reaction, NDM was utilized as the solvent, resulting in coordination with Ag^+^ and Bi^3+^ ions to form a covalent complex as given above (where m and n are positive integers). With the addition of thiourea, these complexes then react with thiourea to form complexes as {Ag[SC(NH_2_)_2_]_m_}^+^ and {Bi[SC(NH_2_)_2_]_n_}^3+^. As the temperature increased, the stability of the complexes decreased to produce AgBiS_2_, which is consistent with the reaction reported by Ganguly et al. [3]

Field emission scanning electron microscopy (FE-SEM) and field emission transmission electron microscopy (TEM) tests were conducted on the as-prepared AgBiS_2_ powders, as depicted in Figure 4 and Figure 5, respectively. The AgBiS_2_ prepared via the reflux method under an ambient atmosphere was granular with an average particle size of about 200 nm, and the particle size and morphology obtained by TEM and SEM were consistent. The lattice fringe labeled in Figure 5d was shown to be 0.325 nm, corresponding to the (111) crystal face of AgBiS_2_. Mak et al. [21] prepared nanocrystals with an average grain size of about 10 nm under room temperature and ambient atmosphere, and Akgul et al. [22] also synthesized nanocrystals with an average grain size of about 5 nm under the same conditions. At room temperature, AgBiS_2_ nanoparticles nucleated but did not grow. The synthesis of the particles shown in Figure 4 was carried out under an ambient atmosphere at elevated temperatures; thus, the AgBiS_2_ nanoparticles under these conditions underwent a grain growth process as compared to the nanoparticles in the above literature.

The thermal stability of the material has a great influence on its performance. Thermogravimetric (TG) analysis was adopted to investigate the thermal stability of the as-prepared AgBiS_2_ powders under an ambient atmosphere. As shown in Figure 6, the black thermogravimetric (TG) curve represents the weight change of the powders, the red derivative thermogravimetric (DTG) curve represents the rate of weight change of the powders, and the blue differential thermal analysis (DTA) curve corresponds to the heat absorption and exothermic change of the powder sample as the temperature rises. It can be seen from the figure that when the temperature rises from room temperature to 700 °C, the weight loss of the powder sample does not exceed 5%; thus, it can be considered that AgBiS_2_ has thermal stability below 700 °C. These results are comparable to previously reported values [23]. The observed weight gain or loss, as well as the absorptive or exothermic processes of the system observed in the curves, occur due to the small number of heterogeneous phases in the powder beyond the accuracy of XRD testing, including the desorption of adsorbed water, the heat absorption process of impurity phase oxidation, and the heat release process of decomposition. All in all, the thermal stability of the AgBiS_2_ powder was determined via the thermogravimetric analysis curve, and its test atmosphere was an air atmosphere, thus confirming that the synthesis of AgBiS_2_ powder could take place in an air atmosphere.

### 3.2. UV–Vis Spectroscopy and Band Gap Characterizations

UV–Vis–NIR absorption spectrum was adopted to describe the optical characteristic of the as-synthesized AgBiS_2_ powders, as shown in Figure 7a. The absorption curve of AgBiS_2_ displays high absorption values in the ultraviolet, visible, and partial infrared regions, and the absorption coefficient decreases gradually when the optical wavelength reaches 1100 nm. As for cubic AgBiS_2_, most reports claim that it is a direct bandgap material [6,17], but some argue that it is an indirect bandgap material [8,24]. Here, the estimated value of bandgap is around 0.98 eV by assuming direct bandgap via plotting (*αhν*)^2^ vs. (*hν*), as shown in Figure 7b. This result is in a good agreement with the reported values of AgBiS_2_ grown via other methods [6,23]. While the AgBiS_2_ powders provide a bandgap of 0.92 eV via the indirect bandgap Tauc plot, their linear fitting relationship is worse than that of the direct bandgap. Therefore, we conclude that the obtained AgBiS_2_ material is a direct semiconductor and features two optical transition mechanisms, with a direct bandgap of 0.98 eV and an indirect band gap of 0.92 eV. These values of optical band gaps are comparable to the previously reported values [23,25]. Adeyemi et al. [23] synthesized AgBiS_2_ and Cu_3_BiS_3_ with a rod-like morphology via a microwave-assisted solution route using a deep eutectic solvent and measured the optical band gap. The black powder of AgBiS_2_ showed an indirect band gap of ∼0.90 eV and a direct band gap of 0.98 eV, which is consistent with our results. Ju et al. [26] investigated the electronic properties of various AgBiS_2_ nanocrystals using first-principles computation and demonstrated that the optoelectronic properties of bulk AgBiS_2_ are highly dependent on the M–S–M–S– (M: Ag or Bi) orderings, and Ag–S–Ag–S– and Bi–S–Bi–S– in AgBiS_2_ bulk crystals contributed, respectively, to the valence band maximum and conduction band minimum. They found that AgBiS_2_ nanocrystals can exhibit markedly different optoelectronic properties depending on their stoichiometry. This may be the main reason for the different band gap values reported.

### 3.3. Photocatalytic Activity

Figure 8 shows the light absorption intensity of the methylene blue solution of the as-prepared AgBiS_2_ samples and the blank control sample after simulated illumination at different times. In the figure, 0 h represents the light absorption intensity of the methylene blue solution at the time when the adsorption–desorption equilibrium is reached under dark conditions and before simulated sunlight irradiation. The peak position of the highest intensity in the absorption peak of the methylene blue solution in the figure is consistent with its characteristic absorption wavelength of 662 nm. Therefore, the change value of its light absorption intensity can be used to represent the change in methylene blue solution concentration to characterize the efficiency of photocatalytic degradation of the methylene blue solution by the catalyst. It is clear that the AgBiS_2_ powder prepared by the reflux method has a certain degradation efficiency within 4 h. However, as to the photocatalytic degradation of the blank control samples, there was almost no spontaneous decomposition of methylene blue under this degradation test condition, so it can be determined that methylene blue underwent photocatalytic degradation under the actions of light and catalysts. The degradation was possible due to the hole, hydroxyl, and superoxide radical, which are the dominant active species generated by the photocatalytic process.

The above data were plotted with time as the abscissa and degradation rate as the ordinate to obtain the photocatalytic degradation efficiency diagram and time as the abscissa and ln(c_0_/c) as the ordinate to obtain the photocatalytic degradation kinetics diagram, as shown in Figure 9. To quantitatively compare the photocatalytic [27] efficiency of all the samples, a pseudo-first-order reaction kinetic model ($ln\left(c_{0}/c\right)=kt+b$, where t is time and b is constant) was employed to calculate the rate constants (k) of the photodegradation experiment over AgBiS_2_ [28]. The degradation of the samples basically occurred via a pseudo-first-order reaction, and the degradation efficiency of the as-prepared AgBiS_2_ sample reached 58.61% with a rate constant of 0.0034 min^−1^, whereas the degradation efficiency of the blank control sample reached 11.10% with a rate constant of 5 × 10^−4^ min^−1^. This indicates that AgBiS_2_ powders prepared via the reflux method have a good degradation effect on methylene blue solution. These degradation results are within the range of previously reported values [3,29]. Ajiboye et al. [29] synthesized graphitic carbon nitride functionalized with ternary silver bismuth sulfide (AgBiS_2_/gC_3_N_4_) for the photocatalytic removal of Pb(II) from water in the presence of methylene blue, crystal violet, and methyl orange. It was observed that the dyes were degraded simultaneously as the divalent lead was reduced to metallic lead, indicating that AgBiS_2_ was a good photocatalyst for removing both dyes and heavy metal ions from water. Ajiboye et al. investigated the degradation rate of dyes in the presence of Pb(II), and the percentage degradation of methyl orange, crystal violet, and methylene blue are 94.85, 71.81, and 39.64%, respectively (corresponding to the pseudo-first-order rate constant of 0.0443, 0.0233, and 0.0104 min^−1^, respectively). The value of 58.61% for the degradation efficiency of the as-prepared AgBiS_2_ powders via the reflux method is better than that of the methylene blue solution reported by Ajiboye et al., while the value of 0.0034 min^−1^ for the degradation rate constant of the methylene blue solution is lower than the reported 0.0104 min^−1^ of the AgBiS_2_/gC_3_N_4_ compounds.

## 4. Conclusions

In summary, a simple and convenient reflux method was adopted to synthesize the completely environmentally friendly AgBiS_2_ powders with a particle size of about 200 nm. The thermal stability of the as-synthesized AgBiS_2_ powders was investigated, revealing that the material retains its integrity up to 700 °C, exhibiting only a 5% weight loss. Furthermore, the AgBiS_2_ powders displayed exceptional optical properties, demonstrating strong absorption across the ultraviolet, visible, and even partial infrared regions. The optical bandgap of AgBiS_2_ was determined to be 0.98 eV, indicating its potential in various optoelectronic applications. The photocatalytic activity of the AgBiS_2_ powders was evaluated through their degradation efficiency of methylene blue solution. Notably, the as-prepared AgBiS_2_ powders exhibited significant degradation efficacy, achieving an impressive degradation efficiency of 58.61% and a rate constant of 0.0034 min^−1^. This highlights the effectiveness of AgBiS_2_ as a catalyst for water pollution control applications. Additionally, the developed reflux method proved to be cost-effective, making it a suitable approach for the large-scale production of AgBiS_2_ powders. Such scalability positions AgBiS_2_ as a promising material for photoelectronic sensing and water pollution control applications, opening up avenues for diverse technological advancements.

## Figures and Tables

**Figure 1 micromachines-14-02211-f001:**
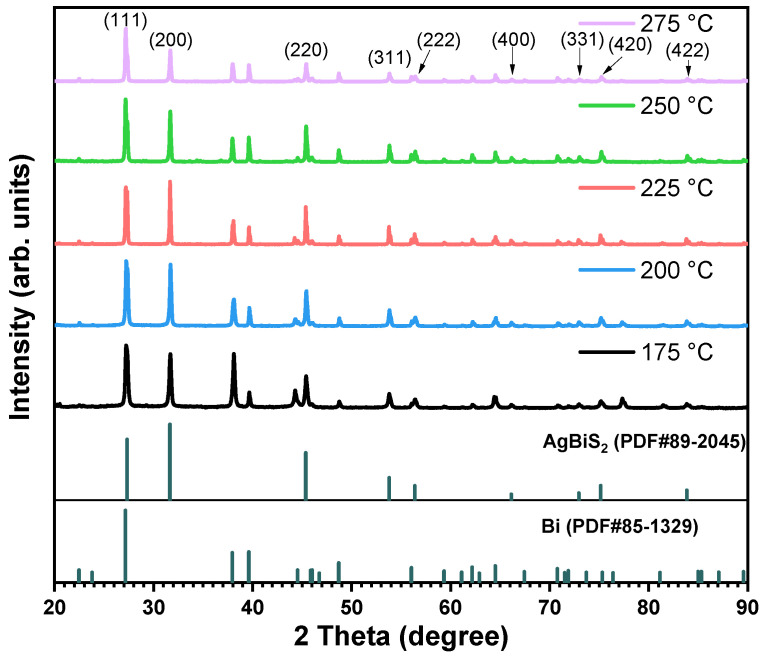
XRD patterns of AgBiS_2_ synthesized at different temperatures with a raw material ratio of 1:1:2.

**Figure 2 micromachines-14-02211-f002:**
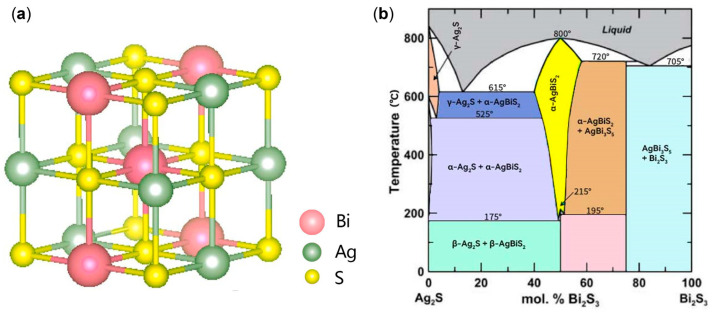
(**a**) Crystal structure of the cubic AgBiS_2_. Gray-green, pink, and yellow spheres denote Ag, Bi, and S atoms, respectively. (**b**) Phase diagram of the Ag_2_S–Bi_2_S_3_ system, modified from [19,20]. Crystal structure: α-Ag_2_S (bcc); β-Ag_2_S (monoclinic); γ-Ag_2_S (fcc); α-AgBiS_2_ (cubic); β-AgBiS_2_ (hexagonal); AgBi_3_S_5_ (monoclinic).

**Figure 3 micromachines-14-02211-f003:**
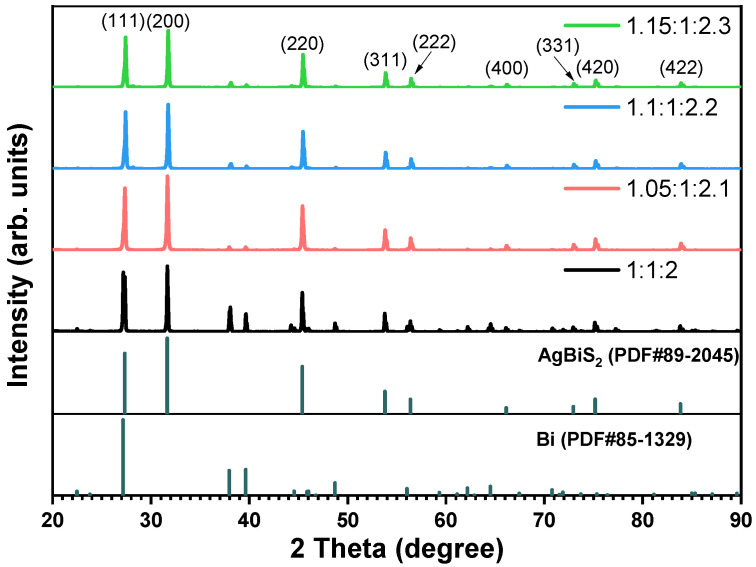
XRD patterns of AgBiS_2_ synthesized with different raw material ratios at 225 °C.

**Figure 4 micromachines-14-02211-f004:**
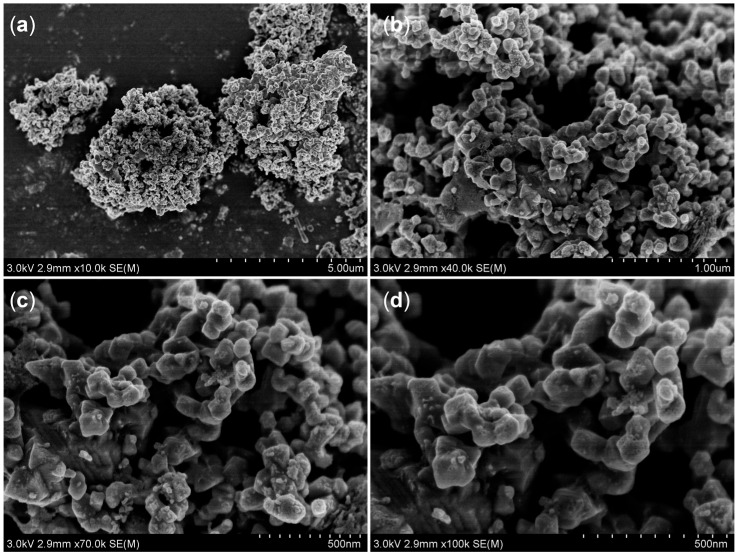
FE-SEM spectrum of AgBiS_2_ powder at 225 °C with a raw material ratio of 1.05:1:2.1 at different magnifications. (**a**) 10 K, (**b**) 40 K, (**c**) 70 K, (**d**) 100 K.

**Figure 5 micromachines-14-02211-f005:**
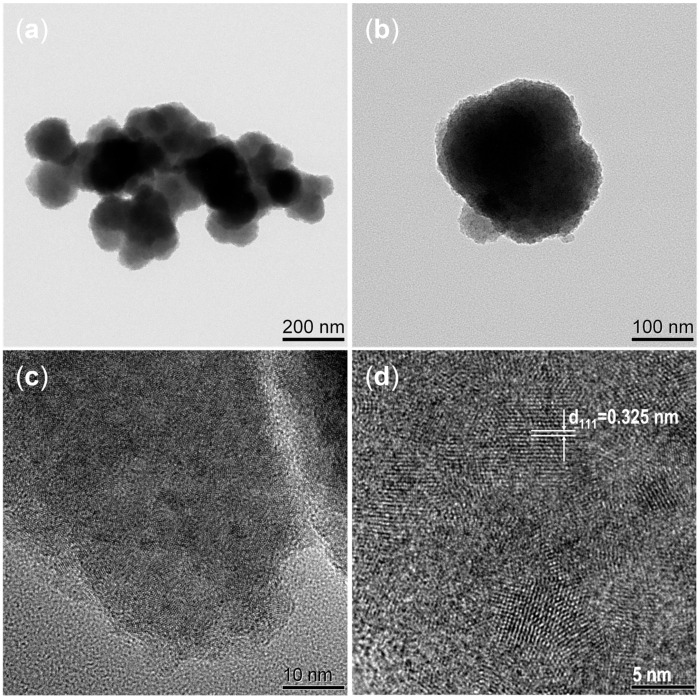
TEM image of AgBiS_2_ prepared via the reflux method in different size bars. (**a**) 200 nm, (**b**) 100 nm, (**c**) 10 nm, (**d**) 5 nm.

**Figure 6 micromachines-14-02211-f006:**
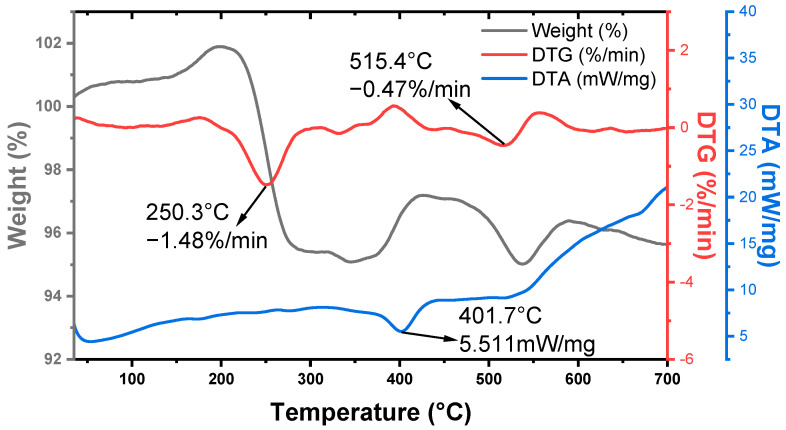
Thermogravimetric analysis curve of AgBiS_2_ powder.

**Figure 7 micromachines-14-02211-f007:**
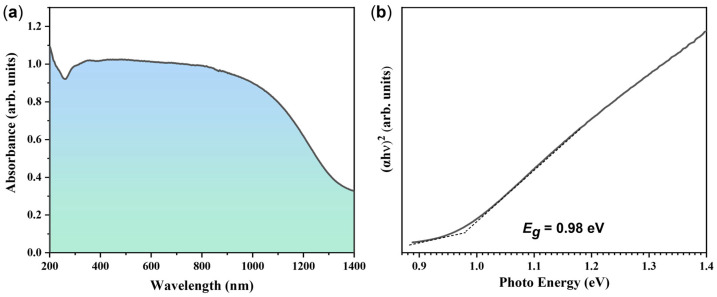
The optical properties of the as-synthesized AgBiS_2_ powder. (**a**) UV–Vis–NIR absorption spectra; (**b**) the corresponding Tauc plots displaying the extrapolated optical band gap.

**Figure 8 micromachines-14-02211-f008:**
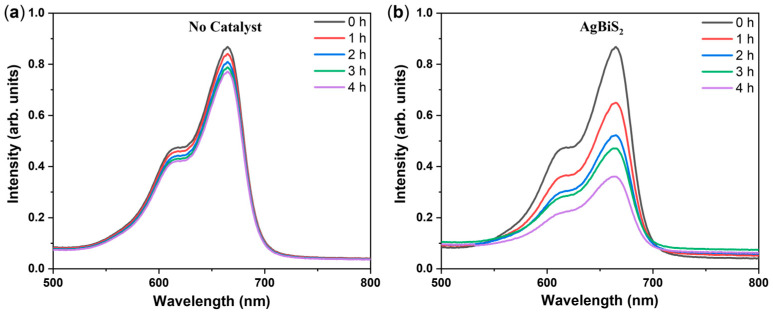
The light absorption intensity of methylene blue solution after photocatalytic degradation of different samples for different times. (**a**) The blank control group; (**b**) AgBiS_2_.

**Figure 9 micromachines-14-02211-f009:**
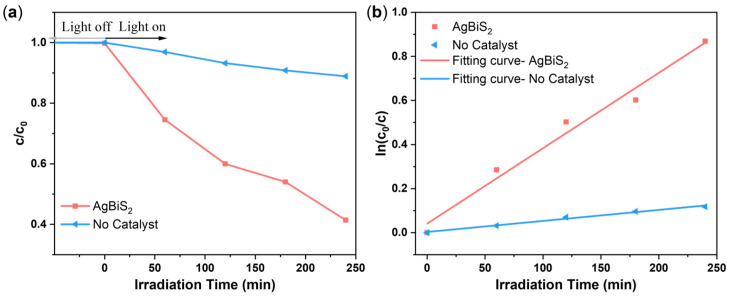
Photocatalytic degradation performance. (**a**) Degradation efficiency; (**b**) kinetics of photocatalytic degradation.

## Data Availability

The data supporting this study’s findings are available from the corresponding author upon reasonable request.

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
