# Peer review of "Cubic AgBiS2 Powder Prepared Using a Facile Reflux Method for Photocatalytic Degradation of Dyes"

_micromachines, 2023, doi:10.3390/mi14122211_

Round 1

Reviewer 1 Report

Comments and Suggestions for Authors

The author reported the preparation and photocatalytic performance of AgBiS2 ,However, the data is not enough , major revision should be considered。 I will offer the authors the opportunity to revise their manuscript.       1 The novelty and research ground of AgBiS2 did not well discussed。                  2 The phase diagram and crystal structure should be provided。      3 the reaction equations are too simple, Tu molecular structure and chemical change should be given and discussed。 Ksp or K complex constant?    4 XRD in Fig 4,there some other peaks which should be discussed。   5  The nanocrystal is solid solution or single phase or heterostructure composed of two phases。 6 the comparison with the previous reports should be listed and discussed。    7 the actived species  in photocatalytic reaction should be added 。   8  the final degradation product should be confirmed。  9 after degradation, what about the XRD and SEM of the catalyst?

Comments on the Quality of English Language

The English should be polished。

Reviewer 2 Report

Comments and Suggestions for Authors

The reported work can be considered for publication after the authors address the following issues:

1.    The author describes the working principles of photocatalysts in the introduction very well. In addition, the introduction needs to include the importance of AgBiS2 and why it is necessary to study or investigate different synthesis methods.

2.    Pg 4 Line 171: Is there a better word to describe ‘book’? Reactor or Teflon flask?

3.    Section 3 (Result) has only 2 lines. Instead, it could be combined with Section 4 as ‘Results and discussion’.

4.    Pg 5 Line 226 “diffraction crystal plane (111), (200), (220) and (311), respectively. , (222), (400), (331), (420), (422).
Need to rephrase/correct this line.

5.    Pg 8 Line 317: The thermogravimetric analysis (TGA) presented does not indicate if AgBiS2 is by reflux or solvothermal method. TGA for both catalysts needs to be provided.

6.    X and Y axes scale range for Fig 13 and 14 should be consistent for better representation.

7.    Need to include a possible explanation for the difference in catalytic activity by different methods. 

Round 2

Reviewer 1 Report

Comments and Suggestions for Authors

The authors stated AgBiS2 powders with an average 16 diameter of 200 nm have been prepared by a simple and convenient reflux method and have a good degradation effect on the methylene blue solution with a degra- 28 dation efficiency of 58.61%。The manuscript is recommended for publication after major revisions.

 1. Given the binary phase diagram of Ag2S-Bi2S3, please.

2. The crystal structure of AgBiS2 should be added.

3. Given the comparison of previous reported the performace of AgBiS2-based photocatalyst.

4. More works provide some useful information about metal sulfide-based nanocrystal materials, growth mechanism and photocatalytic performances such as Chem. Commun., 2015, 51, 1594-1596. Journal of Environmental Sciences 104 (2021) 399–414ï¼›Journal of Environmental Chemical Engineering 8 (2020) 104241; Phys. Chem. Chem. Phys.,2018, 20, 1460-1475

5. The active species  in phtotcatalytic process and band gap structure of the photocatalyst should be discussed.

Comments on the Quality of English Language

The English should be polished.

Round 3

Reviewer 1 Report

Comments and Suggestions for Authors

It seems the authors stated all the comments. I suggested it can be accepted.

Comments on the Quality of English Language

no comments